# Construction of IsoVoc Database for the Authentication of Natural Flavours

**DOI:** 10.3390/foods10071550

**Published:** 2021-07-05

**Authors:** Lidija Strojnik, Jože Hladnik, Nika Cvelbar Weber, Darinka Koron, Matej Stopar, Emil Zlatić, Doris Kokalj, Martin Strojnik, Nives Ogrinc

**Affiliations:** 1Department of Environmental Sciences, Jožef Stefan Institute, 1000 Ljubljana, Slovenia; lidija.strojnik@ijs.si; 2Jožef Stefan International Postgraduate School, 1000 Ljubljana, Slovenia; 3Agricultural Institute of Slovenia, 1000 Ljubljana, Slovenia; joze.hladnik@kis.si (J.H.); nika.weber@kis.si (N.C.W.); darinka.koron@kis.si (D.K.); matej.stopar@kis.si (M.S.); 4Biotechnical Faculty, University of Ljubljana, 1000 Ljubljana, Slovenia; emil.zlatic@bf.uni-lj.si (E.Z.); doris.kokalj@bf.uni-lj.si (D.K.); 5Elaphe, 1000 Ljubljana, Slovenia; martin.strojnik@elaphe-ev.com

**Keywords:** IsoVoc, database, volatile aroma compounds, fruits, headspace-solid phase microextraction (HS-SPME), gas chromatography-combustion-isotope ratio mass spectrometry (GC-C-IRMS), authenticity

## Abstract

Flavour is an important quality trait of food and beverages. As the demand for natural aromas increases and the cost of raw materials go up, so does the potential for economically motivated adulteration. In this study, gas chromatography-combustion-isotope ratio mass spectrometry (GC-C-IRMS) analysis of volatile fruit compounds, sampled using headspace-solid phase microextraction (HS-SPME), is used as a tool to differentiate between synthetic and naturally produced volatile aroma compounds (VOCs). The result is an extensive stable isotope database (IsoVoc—Isotope Volatile organic compounds) consisting of 39 authentic flavour compounds with well-defined origin: apple (148), strawberry (33), raspberry (12), pear (9), blueberry (7), and sour cherry (4) samples. Synthetically derived VOCs (48) were also characterised. Comparing isotope ratios of volatile compounds between distillates and fresh apples and strawberries proved the suitability of using fresh samples to create a database covering the natural variability in δ^13^C values and range of VOCs. In total, 25 aroma compounds were identified and used to test 33 flavoured commercial products to evaluate the usefulness of the IsoVoc database for fruit flavour authenticity studies. The results revealed the possible falsification for several fruit aroma compounds.

## 1. Introduction

Fruits play a significant role in human nutrition. While their consumption depends on several factors, including colour, texture, appearance, and nutritional value, the flavour, which is a combination of aroma and taste, remains the primary selection criteria [1]. The same is valid for fruit-flavoured products where fruity notes continue to play well with all consumers, including those who associate them with health and wellness. The most popular are classic fruit flavours such as apple, berry, and citrus fruits, although flavour preferences are continuously evolving [2]. Most flavourings are produced either by chemical synthesis or by extracting natural materials, although today’s consumers increasingly demand naturally flavoured products [3]. The problem is that the pressure to satisfy consumers and the higher market value of natural flavourings makes naturally flavoured products vulnerable to economically motivated adulteration [4].

New methodologies are being developed to detect food fraud since the analysis of chiral separation of enantiomers, single components, and total aroma spectra have limited applicability [5,6,7]. One such method involves using isotope ratios of light elements such as hydrogen and carbon (^2^H/^1^H, ^13^C/^12^C), which is increasingly being applied in food quality control and in determining the authenticity of natural flavourings. Much effort has also gone into developing isotopic methods to detect adulteration of natural products with synthetic compounds. Among them, gas chromatography-combustion-isotope ratio mass spectrometry (GC-C-IRMS) seems to be the most specific and sophisticated method that can discriminate between natural and synthetic aromas based on the isotopic values of selected volatile organic compounds, VOCs [7,8,9,10,11,12,13,14,15,16,17,18,19,20,21,22,23,24,25]. The use of GC-C-IRMS is the subject of a review by van Leeuwen et al. [8] and a paper by Strojnik et al. [26] in which the authors emphasise the importance of the relevant analytical conditions to obtain precise isotopic ratios.

Articles on determining the authenticity of fruit volatile organic compounds based on GC-C-IRMS are summarised in Appendix A (Table A1). These studies cover, for example, raspberry [10,13,18,19,27], peach [7,11,23,25], strawberry [7,8,20], apple [16,25,28], nectarine [11,23,25], pineapple [7,9], and orange [15,25]. Many aromatic characteristics are shared between different fruits. However, each fruit is characterised by a distinctive aroma that depends on the VOCs present, concentration and perception threshold of individual volatile compounds [29]. Among these aromatic substances, special attention is paid to those compounds that create a fruit’s characteristic aroma and are, therefore, likely to be falsified [29,30]. Studies looking at more than five different VOCs are scarce [7,12,16,20,22], and within those, only two address more than one fruit type [7,16]. In both studies, headspace solid-phase microextraction (HS-SPME) is used to extract the VOCs. The method offers many advantages such as simplicity, reduced time, improved repeatability, lower sample contamination probability [31], and finally, it does not cause isotopic fractionation [26]. HS-SPME-GC-C-IRMS has been used in all four fruit authenticity studies after 2010, proving its suitability as a tool for authenticity verification [7,15,16,32]. Besides fruits, the method has been successfully applied for determining the authenticity of other aromas such as vanilla [31,33,34], citrus essential oils [15,35], wine [36,37,38], and truffle oil [39].

When assessing authenticity, it is essential to have a comprehensive database of authentic samples [16,40,41,42]. The guidelines on the structure and data collection of such a database are presented by Donarski et al. [41]. For example, verification of aroma authenticity is achieved by measuring the isotope values of investigated aroma compounds and comparing the obtained isotopic values with those of reference samples present in the database. Samples containing one or more compounds outside the range established for authentic aroma compounds will be suspected of being adulterated [18]. A limitation of such a database is that it represents only a snapshot in time and requires continuous updating, evaluation, and processing. In addition, using a single fruit type, which is the case in several studies, is insufficient for determining authenticity since different fruit types can have different isotopic ratios [7,8,11]. In addition, all properties that might influence isotopic values must be carefully studied and included in the database. For example, the aroma extraction process must be investigated for evidence of isotopic fractionation, i.e., enrichment of one isotope relative to another. So far, this phenomena has only been addressed by Elss et al. [28] when investigating the technological processing of apple aroma. In this instance, the authors did not observe any isotopic fractionation.

Obtaining all of the necessary distillates of various fruit types to build a database requires extensive resources. However, this task could be significantly reduced by directly analysing the aroma of fresh fruits. Therefore, the present study deals with constructing a stable isotope database (IsoVoc) based on HS-SPME-GC-C-IRMS data to recognise the authenticity of fruit aroma used in food products. Further, the possibility to use fresh fruits instead of distillates for constructing the authentic database on aroma compounds was also investigated. Thus, the specific objectives were to (1) compare *δ*^13^C values of multiple VOCs between distillates and fresh fruit samples (apples and strawberries), (2) compare *δ*^13^C values of VOCs between different fruit types to define the natural variability in δ^13^C values of VOCs, (3) establish a database of *δ*^13^C values of authentic natural and synthetic aroma compounds, and (4) identify aroma compounds as markers of fruit flavour authenticity.

## 2. Materials and Methods

### 2.1. Samples

Fresh apples (n = 86), strawberries (n = 17), raspberries (n = 4), blueberries (n = 3) and pears (n = 2) harvested in Slovenia were obtained from the Agricultural institute of Slovenia and from local food producers, between 2016 and 2019. Additional fresh apples were obtained from the Czech Republic (n = 4), Italy (n = 2), Japan (n = 2), Poland (n = 2), and Slovakia (n = 3). Samples of aroma volatiles, recovered in the water phase (used in text as distillates) in apples (n = 43) and strawberries (n = 17), were prepared by steam distillation at the Biotechnical Faculty, University of Ljubljana. Additional distillates of apples (n = 6), strawberries (n = 2), blueberries (n = 4), raspberries (n = 8), sour cherries (n = 4) and pears (n = 7) with known origin were obtained from a commercial flavour supplier.

Pure synthetically derived aroma compounds were obtained from Sigma Aldrich and from a commercial flavour supplier and include (number in the brackets present the number of obtained samples): **1**, ethyl acetate (n = 2); **2**, ethyl butyrate (n = 2); **3**, ethyl 2-methyl butyrate (n = 2); **4**, butyl acetate (n = 1); **5**, hexanal (n = 3); **6**, 2-methyl butyl acetate (n = 2); **7**, 1-butanol (n = 1); **8**, amyl acetate (n = 1); **9**, butyl butyrate (n = 1); **10**, (E)-2-hexenal (n = 3); **11**, hexyl acetate (n = 1); **12**, (E)-2-hexen-1-ol, acetate (n = 2); **13**, 1-hexanol (n = 1); **14**, (E)-2-hexen-1-ol, (n = 3); **15**, benzaldehyde (n = 2); **16**, hexyl 2-methyl butyrate (n = 1); **17**, methyl butyrate (n = 1); **18**, hexyl butyrate (n = 1); **19**, propyl acetate (n = 1); **20**, methyl acetate (n = 1); **21**, 2-heptanone (n = 1); **22**, 1-octanol (n = 2); **23**, linalool (n = 1); **24**, isopropyl butyrate (n = 1); **25**, propyl propionate (n = 1); **26**, hexanoic acid (n = 1); **27**, isoamyl acetate (n = 3); **28**, methyl hexanoate (n = 2); **29**, acetic acid (n = 1); **30**, E-nerolidol (n = 1) and **31**, ethyl hexanoate (n = 2).

### 2.2. Sample Preparation

Fruit samples were mashed and filtered to obtain fresh juice immediately before analysis. Volatile organic components were extracted from both juice, and the distillates using a Divinylbenzene/Carboxen/Polydimethylsiloxane (DVB/CAR/PDMS) SPME fibre (50/30 µm thickness) purchased from Sigma-Aldrich (Supelco, Bellefonte, PA, USA). The SPME fibre was previously conditioned according to the supplier recommendation. The analysis of the VOCs was performed by sampling the headspace of a 10 mL SPME vial (with silicone/PTFE septa) containing 1 mL of sample. The equilibration time was 10 min at 30 °C, followed by extraction for 20 min at the same temperature. The analytes were then desorbed at 250 °C for 1 min. A working standard solution was prepared by diluting 1 µL of each synthetically derived aroma compound (samples 2 to 15) in 20 mL of water, which was then stored at 4 °C before use. All analyses were performed within three months.

### 2.3. Elemental Analysis-Isotope Ratio Mass Spectrometry (EA-IRMS)

The ^13^C/^12^C ratios were determined using a Vario PYRO Cube analyser (OH/CNS Pyrolyser/Elemental Analyzer) couple to an IsoPrime 100 isotope ratio mass spectrometer, IRMS (IsoPrime, Cheadle, Hulme, UK). The accuracy and precision of measurements were controlled using internal working standards: absolute ethanol (*δ*^13^C = −27.38 ± 0.09‰) and a rum distillate (*δ*^13^C = −13.81 ± 0.09‰) previously calibrated against the certified reference material BCR-656 wine alcohol (*δ*^13^C value = −26.91 ± 0.07‰) from the Institute for Reference Materials and Measurements—IRMM, Belgium).

Carbon isotope ratios are expressed in the *δ*-notation in per mil (‰) relative to the VPDB international standard as follows:*δ*^13^C_VPDB_ (‰) = [(R_sample_ − R_standard_)/R_standard_] × 1000,(1)
where R refers to the ^13^C/^12^C ratios in the sample and standard, respectively. The precision of the measurements was ± 0.1‰.

### 2.4. Gas Chromatography-Mass Spectrometry (GC-MS)

VOCs were identified using a 7890B & 5977A Series GC/MSD (Agilent Technologies, Santa Clara, CA, USA), where chromatographic separation was achieved on a VF-WAXms capillary column (30 m × 0.25 mm × 0.25 µm; Agilent Technologies, Santa Clara, CA, USA). The overall temperature program was as follows: 40 °C (1 min) to 60 °C at 5 °C min^−1^ (held 1 min), then to 100 °C at 7 °C min^−1^, to 180 °C at 10 °C min^−1^ and finally to 200 °C at 15 °C min^−1^ (held 1 min). The carrier gas was helium at a constant flow of 1.5 mL min^−1^. The injection was performed at 250 °C in the split mode (1:10), using a Straight Ultra Inert Liner for SPME (Agilent Technologies, Santa Clara, CA, USA). The ion source was set to 230 °C and the interface temperature to 250 °C, with a scan range of 30 to 400 *m*/*z*. The system was controlled using the ChemStation software (Agilent Technologies, Santa Clara, CA, USA). The identification procedure was performed by comparing retention times and mass spectra with the NIST 14 Mass Spectral Library (Agilent Technologies, Santa Clara, CA, USA) and pure standards.

### 2.5. Gas Chromatography-Combustion-Isotope Ratio Mass Spectrometry (GC-C-IRMS)

Method optimisation and validation are described in detail in Strojnik et al. [26]. Briefly, GC-C-IRMS analysis was performed on Agilent 6890N GC-C system coupled to an IsoPrime GV IRMS. Separation was achieved using an Agilent J&W VF-WAXms capillary column (30 m × 0.25 × 0.25). The overall temperature program was the same as for GC-MS analysis. The carrier gas was also helium at a constant flow of 1.5 mL min^−1^. Injections were performed at 250 °C in splitless mode. The oxidation reactor (Cu/O) in the 6890N GC/C system was set to 900 °C.

Before each measurement, stability and linearity were checked. Acceptable values were < 0.03‰. Reproducibility and accuracy were determined routinely using the working standard. The ^13^C/^12^C ratio was first determined in pure standards and compared with the values obtained by EA-IRMS. Next, the ^13^C/^12^C ratios were determined in the distillates and fruit samples. The multiple-point linear method was used for data normalisation [26]. The reproducibility of the measurements based on duplicate analysis ranged from ±0.1 to ±0.5‰. Finally, the identification of aroma compounds was performed by matching the retention times of pure standards and comparing the chromatograms obtained by GC-MS.

### 2.6. Data Analysis

Metadata (fruit type, fruit variety, processing method, geographical location, and data type, i.e., training or test data) and isotopic values were imported into Excel. Then, data visualisation and data analysis were performed in RStudio. A Mann-Whitney U test for non-normally distributed data was then used to reveal any statistically significant differences. In this case, principal component analysis (PCA) could not be applied since the dataset included many missing values of individual variables.

## 3. Results and Discussion

The primary purpose of creating the IsoVoc stable isotope database for food flavour authentication within the present study is its applicability and usefulness for industrial partners to determine the authenticity of raw ingredients. For this reason, the database must be as broad as possible to cover the widest variety of flavoured fruit products. Therefore, in establishing the IsoVoc database, the following steps were included: selecting authentic reference samples, database creation and authenticity assessment of commercial samples.

### 3.1. Selection of Authentic Reference Sample

To address the usefulness of fresh fruits instead of distillates for database creation, we compared the *δ*^13^C values of specific VOCs present in distillates and fresh fruit samples of apple and strawberry. Only aroma compounds that occur in at least five samples per group (fruit samples and distillates), wherein the compound must be present in both groups, were accepted for further data analysis. In this way, we identified 15 characteristic aroma compounds for apple and 11 aroma compounds for strawberry.

We included 49 recovery aromas and 99 samples of fresh apple juice (Figure 1). For the 15 aroma compounds, *δ*^13^C values ranged between −43.7‰ (acetone in fruit) and −24.1‰ (methyl hexanoate in fruit). A Mann-Whitney U test revealed statistically significant differences in median isotope values for (E)-2-hexanal (−42.2‰; −35.1‰), amyl acetate (−36.5‰; −31.4‰), butyl acetate (−36.2‰; −33.6‰), butyl propionate (−36.8‰; −30.5‰), hexanal (−40.1‰; −35.9‰), hexyl 2-methyl butyrate (−33.9‰; −37.6‰) and hexyl acetate (−36.7‰; −32.3‰) between fruit samples and distillates respectively.

For strawberry, 17 distillates and 17 fresh strawberry samples were analysed (Figure 2). For the selected 11 aroma compounds, *δ*^13^C values ranged between −47.0‰ ((E)-2-hexenal in fruit) and −26.0‰ (2-methylbutyl acetate in distillate). In this case, the Mann-Whitney U test revealed statistically significant differences for (E)-2-hexanal, (−30.5‰; −33.6‰), acetone (−42.1‰; −37.7‰) and ethyl butyrate (−31.4‰; −29.7‰) between fruit samples and distillates, respectively.

Data analysis also revealed several outliers in apple and strawberry that were not a result of analytical problems or random errors but are likely due to differences in variety, stage of maturity, and geographical origin, although further investigation is required. The outliers were not removed from the database. It is also necessary to consider natural isotopic variation caused by, e.g., geographical location, variety, temporal and seasonal variation, and processing to make the database more robust. Further, before data analysis it is important to understand if various technological processes such as distillation cause any isotope fractionation in isotopic values of aroma compounds in specific sample type. When comparing distillates with the freshly prepared fruit juice, we observed a difference in *δ*^13^C values for seven investigated VOCs in apple and three in strawberry. Although Ells et al. [28] did not find a significant isotopic effect caused by technological treatment for (E)-2-hexanal, (E)-2-hexanol, and hexanal, in this study, we found that for (E)-2-hexanal and hexanal, preparing the fresh juice can modify the isotopic data. In apple, the isotopic values for (E)-2-hexanal are lower for fruit juice than distillates. In the case of strawberry aroma, the opposite is true. Except for hexyl-2-methyl butyrate, the median isotope values are lower in the fruit samples. Isotope fractionation that we observed in many volatile compounds is most likely compound dependent. For volatiles that are produced as a result of tissue disruption (secondary compounds, like aldehydes (E)-2-hexanal and hexanal), it seems that different types of fruit processing can cause isotope fractionation in different ways resulting in the difference in isotopic ratios between distillates and fruit juice samples. Beside fruit processing, different fractionation of compounds could be a consequence of flavour changes occurring through maturation, harvest, and subsequent storage. We also observed some differences in isotope values between varieties (such as in red delicious for apples) and samples from different geographical locations (samples from Poland and Japan differentiate from Slovenian ones) that could be important when understanding isotope fractionation rates. However, we cannot draw any general conclusion on this subject since we are dealing with a limited number of samples, and further investigation in this field is required.

Importantly, the following question arises: can fresh fruit samples substitute distillates in the database? For all aroma compounds where the *δ*^13^C values are within the min and max value of the distillates, this is possible, as in the case of 1-hexanol, (E)-2-hexen-1-ol, 2-methylbutyl acetate, ethyl butyrate, and propyl acetate for apples and 1-hexanol, ethyl butyrate, hexyl acetate, methyl butyrate, and methyl hexanoate for strawberries. That means that an authenticity assessment will not be influenced by this effect. However, if we want to cover more fruit VOCs, we need to include both distillates and fruit samples in the database.

Next, for apple and strawberry, distillates and fruits were first combined into a single group (Figure 3). Finally, we compared the data with that for raspberries, blueberries, peaches, pears, and sour cherries to find out which aroma compounds are common to the selected fruit types and which occur most frequently. We then evaluated the *δ*^13^C values of 1-hexanol, (E)-2-hexen-1-ol, (E)-2-hexenal, benzaldehyde, butyl acetate, ethyl acetate, ethyl butyrate, ethyl hexanoate, hexanal, and hexyl acetate. We also found that apples and strawberries account for most of the most variability in the natural range of *δ*^13^C values and contain the highest number of VOCs, which makes them the most appropriate fruits for database creation.

### 3.2. Database Creation

The number of unique VOCs present in at least five samples was 39 (Table 1). Not all variables were present in all samples. Since the dataset included many missing values for individual variables and contained only *δ*^13^C values for authentic natural samples and not for the synthetic samples, we used the Mann-Whitney U test rather than principal component analysis, PCA. We also excluded butyl propionate, isobutyl acetate, estragole, acetone, (Z)-2-hexen-1-ol, acetate, and isopropyl butyrate from Table 1, since there are no data on *δ*^13^C values reported in the literature. However, the *δ*^13^C values for authentic compounds determined in our study are presented, since they could be of a general interest to other researchers and are as follows:: butyl propionate (n = 18; *δ*^13^C from −38.3 to −29.6‰), isobutyl acetate (n = 8; *δ*^13^C from −34.6 to −27.7‰), estragole (n = 34; *δ*^13^C from −44.1 to −36.5‰) in apple, and acetone (n = 23; *δ*^13^C from −43.7 to −35.5‰), (Z)-2-hexen-1-ol, acetate (n = 14; *δ*^13^C from −37.1 to −33.2‰) and isopropyl butyrate (n = 9; *δ*^13^C from −52.9 to −45.1) in strawberry.

All authentic samples were combined in one range (shown in dark green in Figure 4). What is clear is the overlap between individual ranges in *δ^13^C* values. Most data are in good agreement with literature data. For example, Elss et al. [28] reported *δ*^13^C values for trans-2-hexenal ranging from −39.1 to −31.5‰, 1-hexanol ranging from −42.5 to −38.4‰ and trans-2-hexenol ranging from −42.2 to −36.8‰ in apple aroma.

However, for certain VOCs, the authentic range of *δ*^13^C values observed in the literature is slightly shifted (e.g., 1-octanol, (E)-2-hexen-1-ol, hexanoic acid, hexyl butyrate, isobutyl acetate and linalool) compared with this study. For example, isoamyl acetate *δ*^13^C values for bananas are higher, ranging from −29.6 to −27.2‰ [22] compared to our study. This difference likely results from the different preparation method used or technological production, or unique fruit characteristics. In addition, the broad range of *δ*^13^C values in the literature data for γ-decalactone, α-ionone, β-ionone is most likely due to the low number of samples and, consequently, the smaller variation in *δ*^13^C values in samples used in our study. The *δ*^13^C values for γ-decalactone and δ-decalactone in *prunus* fruits such as peaches, apricots, and nectarines ranged from −38.4 to −34.0‰ [11] and were lower compared to strawberries, where the *δ*^13^C values of γ-decalactone ranged from −31 to −28‰ [7]. In this study, the *δ*^13^C values for γ-decalactone in organically produced strawberries range from −29.7 to −28.2‰ and cannot be distinguished from other natural strawberry samples. Raspberry has also been subjected to isotope analysis [13]. The *δ*^13^C values for α-ionone, β-ionone, and α-ionol in these samples ranged from −36.6 to −30.3‰, which shows that it is possible to differentiate between natural and synthetic derived compounds. Although compounds appearing in peach and raspberry are well researched, many more samples are needed to represent better the actual variation in *δ*^13^C values for those compounds.

A database should not contain only the isotopic values of authentic natural samples. For example, we observed several overlaps in the isotopic values of synthetic samples, which meant that it was not possible to differentiate between natural and synthetic samples, mainly (E)-2-hexanal, 2-methylbutyl acetate, amyl acetate, E-nerolidol, ethyl acetate, ethyl hexanoate, hexyl acetate, and linalool *δ*^13^C values. In addition, having only natural aroma compounds in the database is likely to result in samples being misclassified. Different market brands should also be analysed to increase the range of *δ*^13^C values observed in synthetic samples.

Despite this overlap, the method could successfully separate 25 of the 33 VOCs for which we have data for the synthetic compounds (γ-decalactone, α-ionone, β-ionone from the literature).

### 3.3. Authenticity Assessment of Commercial Samples

Based on isotopic mass balance using *δ*^13^C values of individual compounds, estimating the average amount of synthetic compound in the sample is possible, as shown by Strojnik et al. [16]. Here we adapted the method to work for any distribution, not necessarily a normal one, where *δ*^13^C distribution is described by the min, max, and median value. Thus, the *δ*^13^C value of a mixture (*mix*) of two compounds, both natural (*nat*) and synthetic (*syn*), can be expressed with the following relationship:(2)$$\delta^{13}C_{mix}=\left(1-x\right)*\delta^{13}C_{nat}+x*\delta^{13}C_{syn},$$
where fraction *x* corresponds to the added synthetic compound in the mixture, and in the case where *x* = 0.5, both components are present in equal amounts. Assuming an extensive dataset of samples that is representative of the population, we can use $\delta^{13}C_{nat,max}\;and\;\delta^{13}C_{nat,min}$ as the range of *δ^13^*C values in natural compounds (with a median value M) to estimate the lowest value *x* that can be detected for compounds in those regions where natural and synthetic samples are separated. 

When the range of synthetic *δ*^13^C values is higher than the natural range, we use the following expression:(3)$$x=\frac{\delta^{13}C_{nat\;max}-\delta^{13}C_{nat}}{\delta^{13}C_{syn}-\delta^{13}C_{nat}}.$$

For example, we chose benzaldehyde with a median *δ*^13^C*_nat_* value of −31.3‰ and max value $\delta^{13}C_{nat\;max}$ of −28.0‰. If the median *δ*^13^C_*syn*_ value is −25.7‰, then *x* = 0.59, meaning that if a mixed sample contains more than 59% of a synthetic fraction, it is likely (with 50% chance) that adulteration will be suspected since the shift in the *δ*^13^C values will be significant (i.e., out of range of natural compounds). If the calculation is made using the minimum measured values of *δ*^13^C_*nat*_ = *δ*^13^C_*nat min*_ and *δ*^13^C_*syn*_ = *δ*^13^C_*syn min*_, then *x* = 0.79, meaning that if a mixed sample contains more than 79% of a synthetic fraction, the measured *δ*^13^C value will fall out of natural sample range.

If the range of synthetic *δ*^13^C values is lower than natural ones, the corresponding fraction *x* of a synthetic sample is calculated by replacing the maximum values of the measured ranges with the minimum values and vice versa. The detectable fractions of synthetic compounds for different VOCs are presented in Table 2.

The necessary synthetic fraction of selected aroma compounds lies between 5% and 83% for a 50% detection threshold and between 19% and 100% for a 100% detection threshold. Therefore, the method is the most sensitive for detecting adulteration with β-ionone, acetic acid, α-ionone, and 1-butanol and less sensitive at detecting adulteration of ethyl 2-methyl butyrate and hexanal. However, evaluating their usefulness will require a study on how adding different amounts of synthetic aroma compounds to a sample with a weak intrinsic aroma can affect the odour.

Finally, we assessed the authenticity of 33 commercial fruit flavourings, including natural banana, blueberry, peach, grape, pear, apple, strawberry, kiwi, raspberry, blackberry, plum, and sour cherry aroma. A shift in the *δ*^13^C values towards the synthetic range for 2-heptanone, benzaldehyde, butyl acetate, ethyl butyrate, hexanal, hexanoic acid, isoamyl acetate, methyl acetate, methyl hexanoate, and propyl acetate suggest the possible falsification of specific fruit VOCs despite being labelled as natural fruit extracts (Figure 5). Results imply that the authenticity of flavoured products on the market can be questioned, and extensive testing is necessary.

We can also estimate the amount of synthetic fraction x present in the commercial fruit distillates by using the measured $\delta^{13}C_{mix}\;$ values and the minimum, maximum, or average synthetic fraction from the variability of data defining the natural and synthetic database range. By using the median values of VOCs for natural and synthetic samples ($\delta^{13}C_{nat}$ and $\delta^{13}C_{syn}$) we can estimate the median amount of the added synthetic fraction. In the same way, we can also determine the maximum and minimum range of synthetic fraction by using limit database values (minima or maxima) of natural and synthetic samples. Table 3 shows examples for several measured commercial samples that fall outside of the natural range. For example, in the exemplary case of benzaldehyde where the commercial sample has a *δ*^13^C = −26.4‰, using the max value of the synthetic range (−25.3‰) and natural range (−28.0‰), we can estimate that the minimum amount of synthetic fraction is 59%, but it can range up to 95%.

It is impossible to distinguish certain synthetic (petroleum-based) VOCs from natural ones since they can have a *δ*^13^C signature similar to that of a plant [42], which was also observed in our study for (E)-2-hexanal, 2-methylbutyl acetate, amyl acetate, E-nerolidol, ethyl acetate, ethyl hexanoate, hexyl acetate, and linalool. Therefore, developing an HS-SPME-GC-IRMS method for 2D-isotope fingerprinting (^13^C and ^2^H) of VOCs would be needed to prove their natural authenticity.

## 4. Conclusions

The falsification of natural flavours by either dilution, mixing with synthetic compounds, or false declaration of origin of natural ingredients creates a demand to control their authenticity. HS-SPME-GC-C-IRMS analysis of volatile fruit compounds is a valuable tool for differentiating between synthetic and naturally occurring aromas and, together with a database containing flavour compounds with well-defined origins, can be used for authenticity assessment. However, selecting suitable reference samples that need to be analysed to create such a database is critical. When comparing distillates with freshly prepared fruit juice, differences in isotope values in some of the investigated VOCs in apple and strawberry samples is observed. We needed to include both distillates and raw fruit samples to obtain maximal natural stable isotope variability of selected VOCs. A database of *δ*^13^C values for 39 aroma compounds present in fruit and distillate samples (apples, strawberries, blueberries, pears, raspberries, peaches, and sour cherries) and 31 VOC of synthetic origin were established. This study proves that samples of apple and strawberry are sufficient for database construction regarding the variability in *δ^13^C* values. The data are comparable with literature data, and despite some overlap between the natural and synthetic range of values, the method allows the successful separation of 25 of the 33 target VOCs. When testing commercial fruit distillates, the results show possible falsification for several fruit aroma compounds. However, extensive testing of flavoured products on the market is necessary to determine the extent of adulteration and evaluate the usefulness of IsoVoc for different flavoured products. Moreover, the development of an HS-SPME-GC-IRMS method for ^2^H/^1^H ratio determination of fruit VOCs is required to upgrade the existing database, which could be more successfully used in fruit flavour authenticity studies.

## Figures and Tables

**Figure 1 foods-10-01550-f001:**
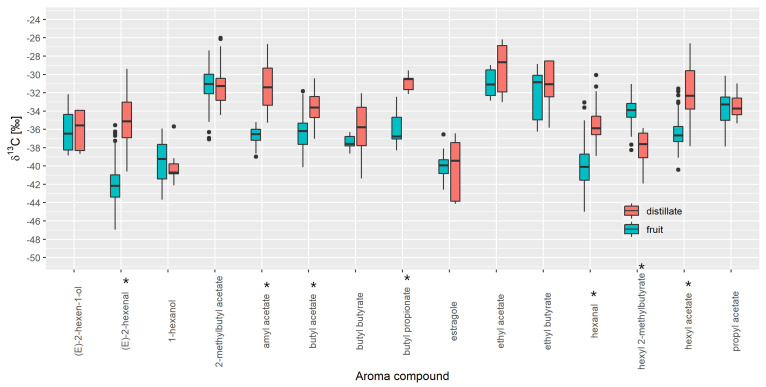
Comparison of *δ*^13^C values of volatile organic compounds (VOCs) present in apple fruit and apple distillates. * *p* < 0.05 (Mann-Whitney test, statistically significant).

**Figure 2 foods-10-01550-f002:**
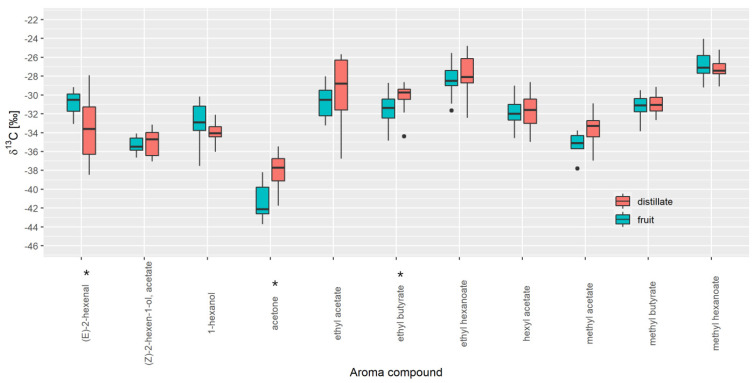
Comparison of *δ*^13^C values of VOCs present in strawberry fruit and distillate. * *p* < 0.05 (Mann-Whitney test).

**Figure 3 foods-10-01550-f003:**
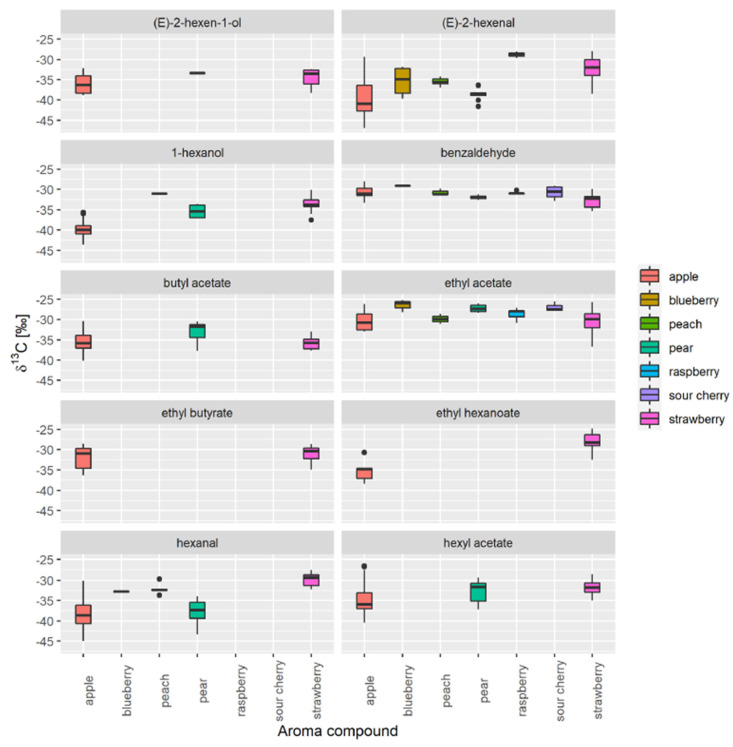
Comparison of *δ*^13^C values of VOCs in different fruit samples. Only aroma compounds present in at least five samples are shown.

**Figure 4 foods-10-01550-f004:**
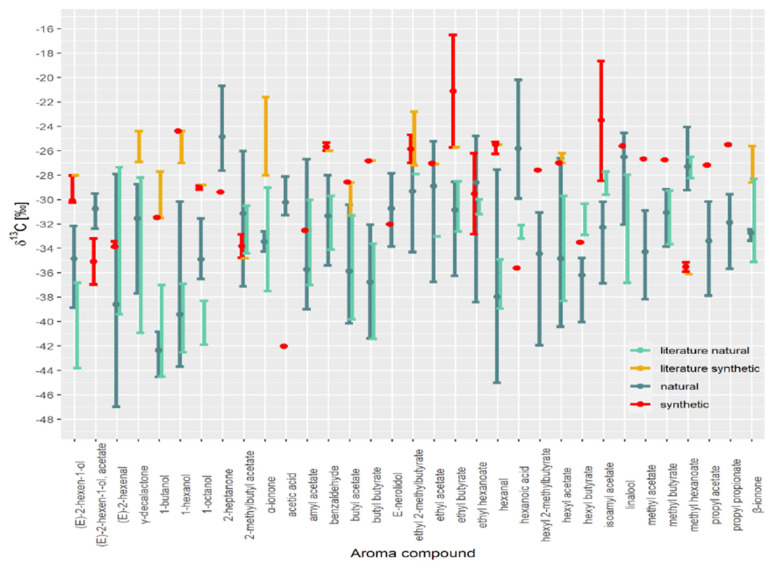
*δ*^13^C values of VOCs of natural and synthetic samples (this study) and literature values.

**Figure 5 foods-10-01550-f005:**
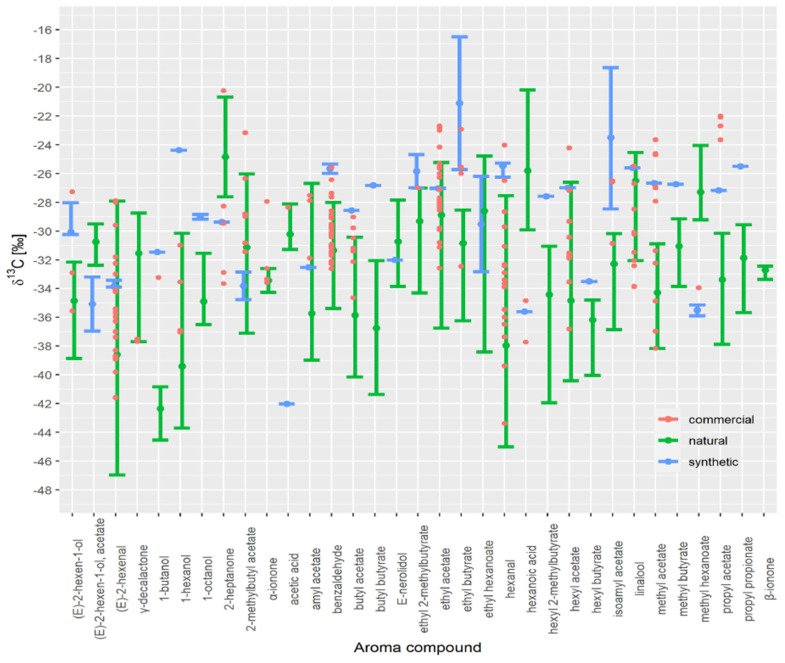
Comparison of *δ*^13^C values of commercial fruit distillates with *δ*^13^C values from the database of natural samples and synthetic standards.

**Table 1 foods-10-01550-t001:** *δ*^13^C (‰) values of volatile organic compounds in synthetic standards, fruits obtained experimentally and from the literature.

	(E)-2-hexen-1-ol	(E)-2-hexen-1-ol, acetate	(E)-2-hexenal	1-butanol	1-hexanol	1-octanol		2-heptanone	2-methylbutyl Acetate	Acetic Acid	Amyl Acetate	Benzaldehyde	Butyl Acetate	Butyl Butyrate	E-nerolidol	ethyl 2-methylbutyrate	Ethyl Acetate	Ethyl Butyrate	Ethyl Hexanoate	Hexanal	Hexanoic Acid	hexyl 2-methylbutyrate	Hexyl Acetate	Hexyl Butyrate	Isoamyl Acetate	Linalool	Methyl Acetate	Methyl Butyrate	Methyl Hexanoate	Propyl Acetate	Propyl Propionate	α-ionone	β-ionone	γ-decalactone
	Synthetic																																	
No.	3	2	3	1	1	2		1	2	1	1	2	1	1	1	2	2	2	2	3	1	1	1	1	3	1	1	1	2	1	1			
min	−30.2	−37.0	−33.9	−31.4	−24.4	−29.2		−29.4	−34.8	−42.0	−32.5	−26.0	−28.6	−26.8	−32.0	−27.0	−27.1	−25.7	−32.8	−26.2	−35.6	−27.6	−27.0	−33.5	−28.5	−25.6	−26.7	−26.7	−35.9	−27.2	−25.5			
max	−28.0	−33.2	−33.4	−31.4	−24.4	−28.8		−29.4	−32.9	−42.0	−32.5	−25.3	−28.6	−26.8	−32.0	−24.7	−27.0	−16.5	−26.2	−25.3	−35.6	−27.6	−27.0	−33.5	−18.6	−25.6	−26.7	−26.7	−35.1	−27.2	−25.5			
	Apple																																	
No.	19	1	127	20	59	4			100		60	33	100	33		15	13	18	6	121		35	102	24	2					25	9			
min	−38.8	−31.9	−47.0	−44.5	−43.7	−33.3			−37.1		−39.0	−33.3	−40.1	−41.4		−34.3	−33.0	−36.2	−38.4	−45.0		−41.9	−40.4	−40.0	−36.9					−37.9	−35.7			
max	−32.2	−31.9	−29.4	−40.8	−35.7	−31.5			−26.0		−26.7	−28.0	−30.4	−32.1		−27.0	−26.2	−28.5	−30.8	−30.1		−31.0	−26.6	−34.8	−36.4					−30.2	−29.6			
	Strawberry																																	
No.	12	10	22		13	7		13	1	5		16	10		5	3	24	29	26	7	6		27	1	3	8	20	29	28					7
min	−38.2	−32.4	−38.5		−37.5	−36.5		−27.6	−33.4	−31.3		−35.4	−37.6		−33.8	−29.1	−36.7	−34.8	−32.4	−32.2	−29.9		−35.0	−35.0	−33.7	−26.9	−37.8	−33.8	−29.2					−35.6
max	−32.4	−29.5	−27.9		−30.2	−34.4		−20.7	−33.4	−28.1		−29.9	−33.0		−27.8	−27.4	−25.7	−28.6	−24.8	−27.5	−20.2		−28.6	−35.0	−30.2	−24.5	−30.9	−29.2	−24.1					−28.7
	Blueberry																																	
No.			5							1		1					3			1						1	4							
min			−39.6							−28.4		−29.1					−28.2			−32.7						−28.5	−38.2							
max			−31.8							−28.4		−29.1					−25.2			−32.7						−28.5	−32.5							
	Pear																																	
No.	1		9		4				3		4	5	7				6			9			6		1									
min	−33.4		−41.6		−37.0				−30.8		−31.9	−32.6	−37.7				−28.4			−43.4			−37.2		−30.9									
max	−33.4		−36.3		−33.5				−27.3		−27.5	−31.2	−30.5				−26.0			−33.9			−29.3		−30.9									
	Raspberry																																	
No.			2						2			5					6									5						8	6	
min			−29.6						−32.7			−31.2					−30.9									−32.0						−34.2	−33.4	
max			−28.0						−31.4			−30.2					−27.1									−30.1						−32.6	−32.4	
	Peach																																	
No.			5		1							3					3			5														2
min			−37.0		−31.0							−31.2					−31.1			−33.7														−37.7
max			−34.2		−31.0							−29.8					−28.6			−29.7														−37.5
	Sour cherry																																	
No.												4.0					3.0									1.0								
min												−32.8					−27.8									−25.5								
max												−29.1					−25.5									−25.5								
	Synthetic literature																																	
min	−28.0			−31.5	−27.0	−28.8			−34.8		−32.5	−26.0	−31.3	−26.8		−27.2	−27.1	−25.7		−25.5			−27.0						−36.1					
max	−28.0			−27.7	−24.4	−28.8			−34.8		−32.5	−26.0	−28.6	−26.8		−22.8	−27.1	−25.7		−25.5			−26.2						−36.1					
	Natural literature																																	
min	−43.8		−39.4	−44.5	−42.5	−41.9			−34.4		−37.0	−34.1	−39.8	−41.4		−27.9	−33.0	−32.6	−31.2	−38.9	−33.2		−38.3	−32.9		−36.8		−33.7	−28.2					
max	−36.8		−27.3	−37.0	−36.9	−38.3			−30.5		−30.0	−29.7	−31.3	−33.6		−27.9	−33.0	−28.5	−30.0	−34.9	−32.1		−29.7	−30.3		−28.0		−29.3	−26.5					

**Table 2 foods-10-01550-t002:** The calculated fractions of added synthetic volatile organic compounds in commercial samples based on the isotope mass balance model.

Aroma Compound	Natural Compound			Synthetic Compound			Detectable Synthetic Fraction	
	Min	Max	Median	Min	Max	Median	x	x
	*δ*^13^C (‰)			*δ*^13^C (‰)			50% Threshold	100% Threshold
γ-decalactone	−37.7	−28.7	−31.5	−26.9	−24.4	−26.1	0.52	0.83
1-butanol	−44.5	−40.8	−42.3	−31.4	−31.4	−31.4	0.14	0.28
1-hexanol	−43.7	−30.2	−39.4	−24.4	−24.4	−24.4	0.61	0.70
1-octanol	−36.5	−31.5	−34.9	−29.2	−28.8	−29.0	0.57	0.68
2-heptanone	−27.6	−20.7	−24.8	−29.4	−29.4	−29.4	0.61	0.80
(E)-2-hexen-1-ol	−38.8	−32.2	−34.9	−30.2	−28.0	−30.1	0.56	0.78
(E)-2-hexen-1-ol, acetate,	−32.4	−29.5	−30.8	−37.0	−33.2	−35.1	0.37	0.78
acetic acid	−31.3	−28.1	−30.2	−42.0	−42.0	−42.0	0.09	0.23
benzaldehyde	−35.4	−28.0	−31.3	−26.0	−25.3	−25.7	0.59	0.79
butyl acetate	−40.1	−30.4	−35.8	−28.6	−28.6	−28.6	0.75	0.84
butyl butyrate	−41.4	−32.1	−36.8	−26.8	−26.8	−26.8	0.47	0.64
ethyl 2-methyl butyrate	−34.3	−27.0	−29.3	−27.0	−24.7	−25.8	0.66	1.00
ethyl butyrate	−36.2	−28.5	−30.9	−25.7	−16.5	−21.1	0.24	0.73
hexanal	−45.0	−27.5	−38.0	−26.2	−25.3	−25.5	0.83	0.93
hexanoic acid	−29.9	−20.2	−25.8	−35.6	−35.6	−35.6	0.42	0.63
hexyl 2-methyl butyrate	−41.9	−31.0	−34.4	−27.6	−27.6	−27.6	0.49	0.76
hexyl butyrate	−40.0	−34.8	−36.2	−33.5	−33.5	−33.5	0.52	0.80
isoamyl acetate	−36.9	−30.2	−32.3	−28.5	−18.6	−23.5	0.24	0.80
methyl acetate	−38.2	−30.9	−34.3	−26.7	−26.7	−26.7	0.45	0.63
methyl butyrate	−33.8	−29.2	−31.1	−26.7	−26.7	−26.7	0.44	0.66
methyl hexanoate	−29.2	−24.1	−27.3	−35.9	−35.1	−35.5	0.23	0.47
propyl acetate	−37.9	−30.2	−33.4	−27.2	−27.2	−27.2	0.52	0.72
propyl propionate	−35.7	−29.6	−31.9	−25.5	−25.5	−25.5	0.36	0.60
α-ionone	−34.2	−32.6	−33.4	−28.0	−21.6	−25.9	0.11	0.26
β-ionone	−33.4	−32.4	−32.7	−28.6	−25.6	−27.7	0.05	0.19

**Table 3 foods-10-01550-t003:** Estimation of the amount of synthetic fraction in commercial fruit distillate.

Aroma Compound	Commercial Fruit Distillate Isotope Value	Min Estimated Synthetic Fraction	Median Estimated Synthetic Fraction	Max Estimated Synthetic Fraction
	*δ*^13^C (‰)	x	x	x
hexanal	−26.5	0.46	0.92	0.99
butyl acetate	−29.8	0.34	0.83	0.89
benzaldehyde	−26.4	0.59	0.86	0.95
ethyl butyrate	−26.0	0.21	0.50	0.97
methyl hexanoate	−34.0	0.71	0.81	0.89
methyl acetate	−27.0	0.92	0.96	0.97
2-heptanone	−28.3	0.38	0.76	0.87
hexanoic acid	−34.8	0.86	0.92	0.95

## Data Availability

No data available.

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
