# Peer review of "Construction of IsoVoc Database for the Authentication of Natural Flavours"

_foods, 2021, doi:10.3390/foods10071550_

Round 1
Reviewer 1 Report
The title of this study is: Construction of IsoVoc database for the authentication of natural flavours.
The purpose of the research was not clearly stated. Write it clearer, please.
I commented on the manuscript and the comments are presented below:
Part 1: Introduction.
The Introduction to the study is too broad and does not end with a clearly stated purpose or goals that the Authors wish to pursue. This should be changed
Part 2: Material and Methods
The Methods section provides the reader with enough information to repeat the experiments conducted. Only the basic statistical analysis was used to determine the differences. With such a large number of parameters tested, which may affect the characteristics examined, the Principal Component Analysis (PCA) should be used to results analyzed. Have the authors attempted to use other more comprehensive statistical analyzes, e.g. principal components analysis of PCA? If yes, please put this method analysis information in this chapter. If not, please explain why? More advanced statistical analysis should be performed.
Part: 3 Results and discussion
For the most part the Results section is well structured.
In the Discussion chapter, there is no comparison and confrontation with the research of other authors in this area from the last years. The results were not fully discussed. A full discussion of the results obtained with other work in this field should be carried out in more aspects. Out of the 34 cited works, only 11 are from the last 10 years, the rest are older.
Part: 4 Conclusion
The Conclusions chapter contains information obtained after conducting experiments but performing only base statistical analyzes.
Part: References.
Out of the 34 cited works, only 11 are from the last 10 years, the rest are older. The literature used is appropriate but should be supplementing about the items from the last years of publication about similar problem. In the references no. 1, 2, 4, 6, 27, 28, 29 and 33 the year should be bold.
Reviewer 2 Report
In my opinion article is very well written and could be accepted as is.
Author Response
See attachement.

Reviewer 3 Report
Dear Authors,
I enjoyed your article and I feel that a database is needed for the aromas. Nevertheless i would like for you to possible elaborate/clarify lines 213-232 of the manuscript. Can you state if there is a fractionation of aroma components originating from distillates and fresh juice? I understand that it is probably component depending, but still can you draw a general conclusion on the distillation versus fresh juices? Lines 269-273 give some other parameters ... these can be excluded for the (E)-2-hexanal and hexanal for which you have different result with the literature?
Author Response
See attachement.

Round 2
Reviewer 1 Report
The authors referred to the comments from the previous review for the manuscript titled: Construction of IsoVoc database for the authentication of natural flavours. I accept explanations. In the future, I suggest using more precise describing relationships between the parameters studied. They supplemented the discussion with a new literature data strengthens the message and importance of information in the manuscript.